# National Survey on Silver Diamine Fluoride (SDF) Awareness, Knowledge, Attitude, and Use among General Dental Practitioners in the Kingdom of Saudi Arabia—An Exploratory Survey

**DOI:** 10.3390/healthcare10112161

**Published:** 2022-10-29

**Authors:** Ali Robaian, Mubashir Baig Mirza, Abdullah Alayad, Malak Almutairi, Ashjan Alotaibi, Alanoud Alroqi

**Affiliations:** 1Department of Conservative Dental Sciences, College of Dentistry, Prince Sattam Bin Abdulaziz University, Alkharj 11942, Saudi Arabia; 2Department of Restorative Dental Sciences, College of Dentistry, King Saud University, Riyadh 11545, Saudi Arabia; 3General Dental Surgeon, Riyadh 11671, Saudi Arabia

**Keywords:** dental caries, dentist, fluoride, knowledge, Saudi Arabia, silver diamine fluoride

## Abstract

Silver diamine fluoride (SDF) has been accepted as an appropriate alternative for caries management. However, knowledge and utilization of SDF among dentists vary considerably. The authors in the present study aimed to assess the awareness, knowledge, attitudes, and use of SDF among general dentists (GD) in Saudi Arabia and to correlate the differences based on the different regions and experience levels of the dentists. In this regard, a cross-sectional web-based questionnaire was conducted, and a response was received from 311 GDs from different parts of the country. Information regarding demographic data, knowledge, attitudes, use, and barriers to SDF in dentists’ professional lives were elicited. The mean age of the participants (55.3%) was between 25–35 years. Most (92.45%) of the dentists were aware of the material and (61%) agreed that SDF could arrest carious lesions. More than half of the dentists agreed/strongly agreed that SDF was a good treatment alternative for restorations in children with behavioral issues (63.1%), medically fragile patients (53.7%), patients with severe anxiety (64.5%), patients who underwent radiation or chemotherapy (47.3%), and patients needing general anesthesia for dental treatment (74%). Comparative evaluation using ANOVA revealed regional differences based on knowledge, attitudes, and use. Tukey HSD further highlighted that the practitioners in the western region are more knowledgeable regarding its benefits and utilize it more frequently in their practice compared to other regions. On the other hand, the experience level of the GDs did not have any impact on their knowledge, attitudes, and use of SDF.

## 1. Introduction

With a greater understanding of the carious disease process and breakthrough advances towards its early diagnosis, the treatment modalities have also undergone tremendous changes in recent years. The traditional methods of treating dental caries using surgical and rehabilitative methods have been challenged, and dental professionals are obliged to consider newer caries management strategies [1,2]. Since dental caries is being recognized as a global public health problem by the World Health Organization [3,4,5,6], clinicians worldwide are exploring newer material possibilities that are minimally invasive, cost-effective, and able to save the time of the dental team and beneficiaries [1].

Dental caries results in demineralization of the inorganic components and destruction of the organic components of the teeth. An array of different materials is constantly being experimentally studied to find materials either capable of reversing this process before the actual cavitation or inhibiting further destruction of sound tooth structure [7]. Silver diamine fluoride (SDF) is one such material that satisfies these characteristics along with its perceived psychological and emotional impacts, especially in anxious patients [8]. Since its introduction in 2015, SDF has been considered an appropriate alternative for caries management [2]. Silver ions have been known for a long time to have good anti-bacterial/anti-enzymatic properties. Moreover, they have an affinity toward the organic components of the dental substrates and readily bind to them [8,9,10]. Fluoride ions, on the other hand, have good calcifying and remineralizing properties after adhering to inorganic components of the tooth structures [11,12]. To take advantage of these properties for managing dental caries non-invasively, research was aimed at combining them to develop a painting material composed of silver and fluoride using amine as a stabilizing agent [13].

Evidence from multiple studies has highlighted SDF’s efficiency as an effective agent in arresting dental caries [7,14], as a relatively inexpensive treatment for the socioeconomically disadvantaged patient groups [15], and as an alternative for patients who cannot tolerate traditional dental care. Furthermore, in primary teeth, SDF use was found to be safe and efficient in arresting caries, which is an added advantage when dealing with high-risk children with intellectual and developmental disabilities [4,14,15]. It reduces the need to perform dental care under general anesthesia, which could result in health risks [16,17]. In the elderly, the use of SDF is also advantageous in preventing and arresting caries, especially on the root surface and in those patients with limited access to dental care. SDF use can help reduce the number of visits required and patients’ anxiety regarding the use of the dental drill and anesthesia. Moreover, SDF can be used to reduce tooth sensitivity [4,13,18].

Although many studies have proven the efficiency of SDF [19,20], studies in Saudi Arabia focused their regional research on the knowledge and attitudes of dentists, including dental students and specialists. Since the majority of the dental workforce in Saudi Arabia consists of general dentists (GD) [21], the authors in the present study aimed to understand the knowledge, awareness, attitude, and use of SDF among GDs in the Kingdom of Saudi Arabia and to correlate the differences amongst them based on different regions and years of experience.

## 2. Materials and Methods

### 2.1. Study Design & Population

This study is a cross-sectional survey conducted among Saudi general dentists practicing in various provinces of Saudi Arabia. Dental students, foreign dentists, and dentists with specializations and board certificates were excluded from the study. The list of registered general dentists and their email addresses was obtained from the Saudi Commission for Health Specialties (SCFHS), Saudi Arabia.

### 2.2. Ethical Considerations

The ethical approval to conduct the study was obtained from the Research Ethics Committee, Prince Sattam bin Abdulaziz University (REC-HSD-106-2021).

### 2.3. Survey

To conduct a nationwide survey, the local governing body, SCFHS, was approached. A further institutional review board ethical approval SRP-000266 was obtained. The recruitment emails were sent to the entire sampling frame of practicing general dentists via the email addresses registered under SCFHS. The subject column of the mail described the aim of the survey to determine the knowledge, attitudes, and professional use of silver diamine fluoride. Furthermore, the main content of the email contained the weblink to an anonymous web-based survey. A mail response within a three-month time period was included. Priori analysis using G* power package 3.1.9.7 was conducted to determine the sample size with an alpha error at 0.05, effect size of 0.15, and at power 0.95. Participants’ responses to the virtual survey were considered implicit consent. A reminder email was sent twice a month.

### 2.4. Questionnaire

The questionnaire was adopted from a previously validated study conducted among American Pediatric dentists [16]. The survey consists of five domains that can be answered as multiple-choice answers and ordinal responses tailored to the particular questions. The first part collected the demographic and general awareness of GD regarding SDF. The second and third parts addressed the professional knowledge of and attitudes towards SDF, respectively. The final two parts measured opinions on the use and barriers to SDF’s use.

Before disseminating the email, a rough survey draft was pilot tested among ten general dentists, and their suggestions were considered. The final survey was reformatted to satisfy the content validity.

### 2.5. Statistical Analysis

The data were imported and analyzed using SPSS, Version 20 (IBM Corp., Armonk, NY, USA). Descriptive statistics were computed to provide an overview of responses using frequencies and percentages along with mean and standard deviation. To determine differences based on different regions and levels of experience, univariate analysis of variance was performed using ANOVA. A Tukey HSD post hoc test was performed to determine the significance of differences between groups.

## 3. Results

In this study, survey responses were received from 311 study participants whose ages ranged from 25 to 50 years. More than half of the practitioners were between the ages of 25–35. Gender-wise distribution revealed an almost equal number of male (51.1%) and female (48.9%) dental practitioners participating in this survey. About 52.45% of the respondents were from the central part of Saudi Arabia, followed by nearly equal proportions from other regions. More than half of the respondents worked in the private sector, and the majority (64.7%) had a practicing experience of 1–5 years.

When asked, “Have you heard about SDF?” 92.45% of the dentist were aware of this material. In descending order, the main sources of information about SDF were online resources, the dental school where they studied, continuing dental education programs, and webinars. However, about 21 respondents were not aware of the SDF material (Table 1).

Regarding SDF knowledge, seven items elicited the respondent’s opinion on a Likert scale. A large majority (61%) agreed or had a firm view that SDF can arrest carious lesions. More than half of the dentists (50.2%) believed that SDF is a good remedy in arresting multiple site carious lesions in a single visit. About two-thirds of the respondents did not agree that SDF should be used before all restorations and an almost equal number had contradictory opinions about SDF usage prior to all restorations in at-risk patients (Table 2).

Table 3 shows dentists’ responses concerning patient-related indications and attitudes regarding SDF use. More than half of the dentists agreed/strongly agreed that SDF was a good treatment alternative for restorations in children with behavioral issues (63.1%), medically fragile patients (53.7%), and patients with severe anxiety (64.5%). Further, they agreed that SDF was a good choice (47.3%) for patients who underwent radiation or chemotherapy and patients who had to be put under general anesthesia for dental treatment (74%).

More than half of the respondents (61.7%) agreed that SDF was a promising treatment alternative for primary teeth but not in the esthetic zone; 47.3% agreed/strongly agreed that SDF is a good alternative for treating permanent teeth, not in the esthetic zone. However, only 22% agreed/strongly agreed with treating primary teeth in the esthetic zone with SDF, and only 18.1% with treating lesions on permanent teeth in the esthetic zone with SDF (Table 3).

The clinicians either disagreed/strongly disagreed that SDF is a good treatment alternative when patients were not able to afford restorative treatment either currently (45.3%) or later (47.6%), as an alternative to general anesthesia treatment (49.2%), and treatment accessing difficulty due to microstomia (49.5%). Notably, most clinicians were neutral on the questions mentioned above (Table 3).

Regarding the frequency of use of SDF in their clinics, about half of the participants reported they had never used SDF in their dental office to prevent carious lesions—51.4% in primary teeth and 62.4% in permanent teeth, respectively. Further, the majority (47.2%) did not use this restorative for tooth sensitivity; 35.7% used it either sometimes or often. When asked about their future use of SDF, 26.4% reported it would increase a little, and only 0.6% thought it would increase significantly (Table 4).

Table 5 revealed, in ascending order, that the barriers to SDF use were staining on teeth (52.1%), improper tooth contour (40.5%), patient acceptance (33.1%), and cost (28%). The mean average scores for knowledge, attitude, usage, and barriers to SDF usage were 3.2, 2.8, 2.2, and 2.8, respectively.

Analysis of variance (ANOVA) was used for regional comparison among general dentists in Saudi Arabia based on their levels of knowledge, attitudes, and use, which revealed highly significant differences as seen in Table 6.

Tukey HSD test was further performed to find the means of which specific regions are significant (Table 7). When knowledge of the GDs in different regions were compared with each other, those based in the west had more information about SDF than other groups and showed statistically significant differences compared to those in the southern and northern regions. A significant difference was also seen among GDs in the northern and central regions. No significant difference was seen among GDs when comparing between the other regions (Table 7).

With regard to considerations/attitudes, statistically significant differences were seen among the GDs from eastern, western, and central regions compared to those from the north. However, there was no difference between the GDs from the north and south. A significant difference was also seen in GDs of the western region when compared to the southern region (Table 7).

Regarding the use of SDF, the mean scores of GDs in the central region showed the lowest use compared to other regions, with a statistically significant difference seen with western regions. Higher use of SDF was seen among GDs from the west, which was also statistically significant compared to the northern and southern regions as seen in Table 7.

ANOVA was used to compare knowledge, attitudes, and use among GDs based on different levels of experience as seen in Table 8. The data show that the means for knowledge and attitudes were higher in fresh graduates with less than two years of experience. However, the results were not significant, although use was higher among GDs with an average experience ranging from six to ten years.

## 4. Discussion

Dental caries is one of the most prevalent chronic diseases that does not follow the inverse care law [1,3,22,23,24]. In Saudi Arabia, it is estimated to be prevalent in 80% of children, affecting both primary and permanent dentition [25,26]. Treatments aimed at prevention, primarily targeting inhibition of caries progression, are a viable method to control this condition [8,27,28]. The profession has gradually shifted from the paradigm of extension of the cavity for prevention to the concepts of minimal intervention, including first occurrence, earliest detection, preventive interception, and minimally invasive patient-friendly treatment [29]. The use of SDF is one such non-invasive method to manage dental caries either at the incipient stage or to treat a cavitated lesion, preventing further destruction. Its procedure requires a very short time application of inexpensive materials [30]. Many studies, clinical trials, and systematic reviews demonstrate that the application of SDF arrests or stops the progression of carious lesions in a high percentage of cases (30–70%) [31,32,33].

To our knowledge, only one study among GDs in the Hail region of Saudi Arabia reported comprehensive knowledge of SDF [1]. Conflicting views about the knowledge, efficacy, and clinical application of the newly introduced materials in general practitioners [4] motivated the authors of this study to assess the domains mentioned above among general practitioners across the region for generalizability. In the present study, more than half of the respondents were under 35 years of age with a maximum of five years of clinical experience. This may be because younger generations are more comfortable with an online survey, or the participants who are familiar with the topic tend to reply to the survey, which may cause response bias. It could also be because of the lower data in this study due to a poor response rate by the respondents. However, it is important to note that gender bias was addressed by the almost equal distribution of both genders. Furthermore, almost equal number of general dentists working in the government and the private sector participated in our work.

Mirroring the results of the present study (60%), general practitioners in the Riyadh region [4] also agreed that SDF could arrest cavitated lesions. In contrast, in a similar survey conducted among Japanese dentists, about 90% considered it an effective tool against dental caries [7]. The higher acceptance of this material in the Japanese survey could be because the respondents were both general dentists and specialists. Higher knowledge among specialist dentists was previously recorded in American pediatric dentists [16]. Moreover, Japan was the first country to introduce SDF for dental treatments. The other reason could be the data obtained in our study were considerably lower.

Although the American Academy of Pediatric Dentists (AAPD) chairside guidelines for the use of SDF states that it is not necessary to remove the carious dentin before SDF application [30], more than half of the participants in our study did not agree on this step, similar to a study among pediatric specialists [16]. Similarly, SDF was first introduced as a means to relieve dentinal hypersensitivity, with evidence from several studies suggesting a high success rate [7,34]. However, many practitioners are oblivious to this use of SDF, as suggested by the results of this study. These observations among the general practitioners in the Kingdom indicate that knowledge about its clinical use needs further updating through necessary interventions or programs.

Concerning attitudes/conditions related to the usage of SDF, the participants in our study had a similar perception to those seen in another study in the Riyadh region of Saudi Arabia [4]. However, in that study, the GDs were more inclined to use SDF in treating anxious patients and as an alternative to general anesthesia. The mean scores achieved in this domain were comparatively lower than those achieved among pediatric dentists in the United States [16].

The effectiveness of SDF in arresting dental caries is known to be up to 47–90% and is much higher in the anterior teeth [30,35]. Only half of the general dentists in our study responded positively regarding its use in anterior teeth, which fall in the esthetic zone. Surprisingly, more than two-thirds of the pediatric dentists in the USA responded that SDF could not be used for restoring dental caries in the esthetic zone for primary or permanent teeth [16]. Considering teeth in the non-esthetic zone, many dentists were neutral in their responses. It is surprising to note that, although most specialists or general dentists know about SDF, their attitudes towards its usage vary considerably. Hence, it is essential to study the information on the percentage of respondents who answered incorrectly or remained neutral for planning future research and educational efforts.

Among young preschool children, Early Childhood Caries (ECC) is prominently seen, and in children with special needs, restorative care is always challenging, and the child usually becomes restless, which necessitates the use of either moderate sedation or general anesthesia [36]. SDF seems to be a promising alternative in treating such patients with a high level of acceptance among their parents/caregivers [37]. Results from our study also show the respondents’ inclination towards its use in such patients. Moreover, evidence also indicates that SDF gives more promising results in primary teeth [2,18,38]. As for the barriers to use the SDF, black discoloration on the tooth was stated to be a major barrier. However, among the general practitioners of Riyadh city, cost was the main obstacle to its use, which is surprising given the fact that SDF is a cheaper treatment option [4]. This response could have been due to improper knowledge and lack of previous use of the material.

However, this study has a number of limitations, including the poor response rate: the response rates of web-based and emailed surveys are usually low, which partly explains the response in this article. Moreover, there are higher chances of bias, as the respondents who are more interested in a particular topic tend to respond more frequently to such surveys.

## 5. Conclusions

Under the limitations of this study, it can be concluded that the awareness of SDF among GDs in Saudi Arabia is high, and a majority of this awareness is attained from online resources. Mean scores about knowledge and attitudes were higher; however, the barriers to use could have resulted in less usage of SDF. The GDs in the western part of Saudi Arabia were more knowledgeable, and use SDF more frequently, when compared to GDs from other regions. However, the experience levels of the GDs did not influence any of the tested parameters.

## Figures and Tables

**Table 1 healthcare-10-02161-t001:** Characteristics of general dentists participating in the survey about silver diamine fluoride (SDF).

Age	Frequency (n)	Percent (%)
25–35 years	172	55.3
36–50 years	102	32.8
Above 50 years	37	11.9
Gender		
Male	159	51.1
Female	152	48.9
Region		
Central	163	52.4
East	48	15.4
North	35	11.3
South	38	12.2
West	27	8.7
Current workplace		
Government sector	143	46.0
Private sector	168	54.0
Years in Practice		
Less than 2 years	110	35.4
2–5 years	91	29.3
6–10 years	61	19.6
Greater than 10 years	49	15.8
Have you heard about Silver Diamine Fluoride (SDF) application in Dentistry?		
Yes	288	92.6
No	23	7.4
How did you hear about (SDF)?		
Continuing education programs	70	22.5
In the dental school	84	27.0
Not applicable	21	6.8
Online resources	93	29.9
Webinars/seminars	43	13.8

**Table 2 healthcare-10-02161-t002:** Participating dentist responses about their silver diamine fluoride (SDF) knowledge, by the percentage of respondents to each item.

	1	2	3	4	5	Mean	SD
SDF can be used to arrest cavitated lesions	-	10.9%	28%	35.3%	25.7%	3.75	0.958
SDF can be used to arrest non cavitated lesions	-	14.5%	32.2%	32.2%	21.2%	2.99	1.159
Infected soft dentin must be removed prior to applying SDF	11.9%	21.9%	31.5%	24.8%	10%	2.99	1.159
SDF is a good treatment for arresting caries when it is not possible to restore all lesions in one appointment	5.8%	10%	34.1%	29.3%	20.9%	3.49	1.103
SDF should be used prior to placing all restorations in all patients	18.6%	25.7%	33.1%	16.7%	5.8%	2.65	1.133
SDF should be used prior to placing all restorations in at-risk patients	-	14.5%	33.8%	34.1%	17.7%	3.54	0.945
Average Score						3.23	0.4208

Response options were 1 = strongly disagree, 2 = disagree, 3 = neutral, 4 = agree, and 5 = strongly agree.

**Table 3 healthcare-10-02161-t003:** Participating dentist responses regarding silver diamine fluoride (SDF) considerations/attitudes, by the percentage of respondents to each item.

SDF Is a Good Treatment Option for Lesions That Are:	1	2	3	4	5	Mean	SD
In the esthetic zone on primary teeth	19%	30.2%	28.6%	16.4%	5.8%	2.59	1.139
Not in the esthetic zone on primary teeth	-	10%	28.3%	34.7%	27%	2.38	1.132
In the esthetic zone on permanent teeth	26.4%	30.2%	25.4%	14.2%	3.9%	2.70	1.202
Not in the esthetic zone on permanent teeth	-	19.9%	32.8%	26.7%	20.6%	2.65	1.08
For restorations in children with behavioral issues	-	5.8%	31.2%	35.1%	28%	2.73	1.208
When patients have severe dental anxiety	-	8.4%	27%	41.4%	23.2%	2.86	1.188
When patients are undergoing or have recently undergone radiation therapy or chemotherapy	3.5%	9.3%	39.9%	34.1%	13.2%	3.44	0.954
When patients take bisphosphonate medications	-	9.6%	44.7%	29.2%	16.4%	2.86	1.015
When a patient wants to place a restoration at a later time as he cannot currently afford it	-	13.8%	33.8%	38.2%	14.1%	2.96	1.043
When patients cannot pay for restorations	-	13.5%	31.8%	37%	17.7%	2.88	1.096
If patients would have to be put under general anesthesia for dental treatment	-	16.1%	33.1%	32.8%	18%	2.80	1.084
If patients would be unable to receive normal dental treatment and could also not be put under general anesthesia for treatment	2.6%	9.3%	34.4%	37.9%	15.8%	3.54	0.952
If patients with microstomia have difficulty accessing lesions that require treatment	-	11.6%	37.9%	39.2%	11.3%	3.05	0.979
Average Score						2.88	0.3206

Response options were 1 = strongly disagree, 2 = disagree, 3 = neutral, 4 = agree, and 5 = strongly agree.

**Table 4 healthcare-10-02161-t004:** Participating dentists’ responses about their use of silver diamine fluoride (SDF), by percentage of respondents to each item.

Use of SDF	1	2	3	4	5	Mean	SD
How often did/do you use SDF in your office to treat tooth sensitivity	47.2%	17%	24.4%	6.8%	4.5%	2.04	1.181
How often did/do you use SDF in your office to prevent dental caries	51.2%	14.1%	22.5%	10%	1.3%	1.94	1.123
How often did/do you use SDF in your office to arrest dental caries in primary teeth	51.4%	13.8%	20.3%	10%	4.5%	2.02	1.235
How often did/do you use SDF in your office to arrest dental caries in permanent teeth	62.4%	9.6%	18.6%	7.4%	1.9%	1.76	1.109
Do you expect your future usage of SDF to ^b^	13.2%	16%	16.7%	26.4%	0.6%	3.28	1.458
Average Score						2.20	0.6093

Response options were 1 = never, 2 = rarely, 3 = sometimes, 4 = often, and 5 = very often. ^b^ Response options were 1 = decrease a lot, 2 = decrease a little, 3 = not change, 4 = increase a little, and 5 = increase a lot.

**Table 5 healthcare-10-02161-t005:** Participating dentists’ responses about barriers of silver diamine fluoride (SDF), by the percentage of respondents to each item.

	1	2	3	4	Mean	SD
Leave a tooth without proper anatomy if not restored	12.5%	28%	18.6%	40.8%	2.87	1.084
A permanent dark mark on the tooth	30.2%	21.9%	17.4%	30.5%	2.48	1.212
Patients/caregivers acceptance of the treatment.	10.9%	22.2%	22.2%	44.7%	3.06	1.053
Cost of SDF	5.8%	22.2%	41.8%	30.2%	2.96	0.869
Average Score					2.84	0.25382

Response options were 1 = Extreme barrier, 2 = Moderate barrier, 3 = Not a barrier, 4 = Somewhat a barrier.

**Table 6 healthcare-10-02161-t006:** Regional comparison of variables among general dentists about silver diamine fluoride (SDF).

	Region	N	Mean	Std. Deviation	Minimum	Maximum	ANOVA	
							F	*p*
Knowledge	North	35	17.286	7.274	6.000	28.000	5.108	0.001 **
	South	34	18.059	3.733	12.000	23.000		
	East	46	18.957	2.781	13.000	27.000		
	West	23	21.217	3.261	13.000	26.000		
	Central	149	19.839	3.369	13.000	30.000		
Attitude	North	29	35.586	4.642	13.000	56.000	7.706	0.000 **
	South	34	38.618	7.307	26.000	51.000		
	East	42	41.857	5.462	28.000	52.000		
	West	25	45.880	5.826	35.000	54.000		
	Central	147	42.633	7.200	27.000	65.000		
Use	North	33	11.273	5.496	5.000	22.000	7.696	0.000 **
	South	34	11.324	4.183	5.000	21.000		
	East	46	12.478	4.010	5.000	19.000		
	West	25	15.520	4.254	9.000	21.000		
	Central	147	10.388	4.549	5.000	25.000		

F, F value in ANOVA; *p*, *p*-value; ** highly significant with *p*-value ≤ 0.05.

**Table 7 healthcare-10-02161-t007:** Multiple comparisons using Tukey HSD for regional differences among variables.

	Region		Mean Difference	*p*	95% Confidence Interval	
					Lower Bound	Upper Bound
Knowledge	North	South	−0.773	0.930	−3.420	1.874
		East	−1.671	0.341	−4.137	0.795
		West	−3.932	0.003 *	−6.882	−0.981
		Central	−2.553	0.007 *	−4.618	−0.488
	South	East	−0.898	0.859	−3.384	1.589
		West	−3.159	0.031 *	−6.126	−0.191
		Central	−1.780	0.136	−3.869	0.309
	East	West	−2.261	0.179	−5.068	0.546
		Central	−0.882	0.687	−2.737	0.972
	West	Central	1.378	0.539	−1.084	3.841
Attitude	North	South	−3.031	0.563	−8.579	2.516
		East	−6.271	0.011 *	−11.570	−0.972
		West	−10.294	0.000 *	−16.283	−4.304
		Central	−7.046	0.000 *	−11.506	−2.587
	South	East	−3.239	0.401	−8.302	1.823
		West	−7.262	0.006 *	−13.044	−1.480
		Central	−4.015	0.066	−8.191	0.161
	East	West	−4.023	0.272	−9.567	1.521
		Central	−0.776	0.981	−4.615	3.064
	West	Central	3.247	0.332	−1.500	7.995
Use	North	South	−0.051	1.000	−3.084	2.982
		East	−1.206	0.769	−4.037	1.626
		West	−4.247	0.004 *	−7.538	−0.956
		Central	0.885	0.848	−1.506	3.276
	South	East	−1.155	0.791	−3.962	1.653
		West	−4.196	0.004 *	−7.467	−0.926
		Central	0.936	0.813	−1.426	3.298
	East	West	−3.042	0.055	−6.126	0.042
		Central	2.091	0.051	−0.007	4.188
	West	Central	5.132	0.000 *	2.447	7.818

*p*, *p*-value; * significant with *p* ≤ 0.05.

**Table 8 healthcare-10-02161-t008:** Comparison of variables based on experience among general dentist about silver diamine fluoride (SDF).

	Experience	N	Mean	Std. Deviation	Minimum	Maximum	F	*p*
Knowledge	<2 years	96	19.7708	4.70941	6.00	30.00		
	2–5 years	87	18.6552	3.50026	6.00	24.00	1.266	0.286 ns
	6–10 years	57	19.1404	4.60760	6.00	27.00		
	>10 years	47	19.6383	3.07462	14.00	28.00		
Attitude	<2 years	90	42.2111	9.82829	13.00	65.00		
	2–5 years	85	40.6353	7.67471	13.00	55.00	0.573	0.633 ns
	6–10 years	57	41.9825	8.49683	25.00	56.00		
	>10 years	45	41.5778	6.08832	29.00	52.00		
Use	<2 years	94	10.9681	5.24548	5.00	25.00		
	2–5 years	85	11.3765	4.19176	5.00	21.00	0.819	0.485 ns
	6–10 years	59	12.1864	4.48167	6.00	22.00		
	>10 years	47	11.2553	4.88328	5.00	21.00		

*p*, *p*-value; ns, non-significant with *p*-value at ≤ 0.05.

## Data Availability

The datasets used and/or analyzed during the current study are available from the corresponding author upon reasonable request.

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
