# Peer review of "National Survey on Silver Diamine Fluoride (SDF) Awareness, Knowledge, Attitude, and Use among General Dental Practitioners in the Kingdom of Saudi Arabia—An Exploratory Survey"

_healthcare, 2022, doi:10.3390/healthcare10112161_

Round 1

Reviewer 1 Report

First of all, I would like to congratulate the authors for this research work, but I sent them some questions that need to be answered.

1. The following sections should not be numbered in the abstract: (1) Background.

2.In points 2.3 and 2.4 of the article, it talks about the survey used, but does not refer to any article that has previously used it and the validation of the same.

3. The results only show descriptive data, but no multivariate analysis has been performed. For example, you can see if there are differences between men and women or between years of experience. Justify in a scientific way why you have not performed an in-depth statistical analysis.

4.In line 188 to 193 in the discussion he talks about differences between years of experience, however, he does not demonstrate that these years are so because the data obtained are not sufficient.

5.In the discussion he does not mention the limitations of the study such as points 4 and 3 that I have explained to him previously.

6. The conclusions cannot be sustained with the results that it shows, if they are a consequence of them but they are very general.

Reviewer 2 Report

Manuscript title

National survey on Silver Diamine Fluoride (SDF) awareness, 2 knowledge, attitude, and use among General Dental practition- 3 ers in the Kingdom of Saudi Arabia-A exploratory survey

The authors present results from a study conducted in the Kingdom of Saudi Arabia Romania to assess the awareness, knowledge, attitude, and use of SDF among general dental practitioners in Saudi Arabia. Although the research topic and the source of data are relevant, several aspects need to be improved or clarified in the manuscript. The main shortcomings are: the lack of a clear rationale for the study, the lack of a clear description of the methods, lack of analysis of the metric characteristics of the instrument, which is the greatest weakness of the paper. In general, the authors should carry out a step prior to the application of an instrument, its validation.

Abstract

In general, the abstract is well balanced.

1. As far as my information, the policy of MDPI is to have non-structured abstract. please make the change.

2.  Delete the numbers that the authors placed before the headings.

3. Specify in the methods section that the participants were general practice dentists.

4. Consider mentioning the age range and mean age of the participants.

5. Mention something about the statistical analysis performed.

6. The conclusions do not agree with the objective of the study. The authors mention "awareness, knowledge, attitude and use" and it is on these four issues that they must conclude.

Introduction

7. What is the research question and what is the hypothesis? lack of a clear rationale for the study.

8. The objective of the study is not the same as that of the abstract. It is necessary to place what is related to "use"

Material and Methods

In general, the methodology needs to be described in greater detail to be accepted

9. Please provide additional information on exclusion criteria.

10. How did the authors arrive at the sample? Was there any sample size calculation?

11. What did the authors measure about attitudes? It is not mentioned in the description of the questionnaire.

13. Being a development of the questionnaire, the authors must carry out some analysis of the metric characteristics of the instrument such as the evaluation of viability, reliability, validity and sensitivity to change.

Results

14. The results are purely descriptive, the authors do not present any metrics of the instrument, which is mandatory when developing an instrument. Since the authors did not (mention) used previously validated instruments. which is the main limitation of the study.

Discussion

15. Given the lack of rationality of the study and the lack of measures of the characteristics of the instrument, the discussion cannot be evaluated. The section needs organization, they have too many paragraphs, most of which are a couple of lines long. They seem isolated ideas.

16. Authors should have a limitations section. For example, discuss the selection bias, sample size, sample selection, etc.

Conclusion

17. The authors say nothing about their results. Conclusions shown are not conclusions derived from their results. It is necessary to add conclusions based on the results of your study. Reconsider rewriting the conclusion based on the current study and research results.

Round 2

Reviewer 1 Report

The manuscript has been significantly improved.

In my opinion it can be published in the journal.

Best Regards

Reviewer 2 Report

The authors made the changes suggested by me.